# Mapping the discursive dimensions of the reproducibility crisis: A mixed methods analysis

**Nicole C. Nelson**[1]*, **Kelsey Ichikawa**[2], **Julie Chung**[2], **Momin M. Malik**[3]

**1** Department of Medical History and Bioethics, University of Wisconsin, Madison, Wisconsin, United States of America, **2** Radcliffe Institute for Advanced Study, Harvard University, Cambridge, Massachusetts, United States of America, **3** Berkman Klein Center for Internet & Society, Harvard University, Cambridge, Massachusetts, United States of America

* nicole.nelson@wisc.edu

**Data Availability Statement:** Complete bibliographic information for the data set is available at: https://www.zotero.org/groups/2532824/reproducibility-ca. The data files exported from NVivo and the code book (which provides the

## Abstract

To those involved in discussions about rigor, reproducibility, and replication in science, conversation about the "reproducibility crisis" appear ill-structured. Seemingly very different issues concerning the purity of reagents, accessibility of computational code, or misaligned incentives in academic research writ large are all collected up under this label. Prior work has attempted to address this problem by creating analytical definitions of reproducibility. We take a novel empirical, mixed methods approach to understanding variation in reproducibility discussions, using a combination of grounded theory and correspondence analysis to examine how a variety of authors narrate the story of the reproducibility crisis. Contrary to expectations, this analysis demonstrates that there is a clear thematic core to reproducibility discussions, centered on the incentive structure of science, the transparency of methods and data, and the need to reform academic publishing. However, we also identify three clusters of discussion that are distinct from the main body of articles: one focused on reagents, another on statistical methods, and a final cluster focused on the heterogeneity of the natural world. Although there are discursive differences between scientific and popular articles, we find no strong differences in how scientists and journalists write about the reproducibility crisis. Our findings demonstrate the value of using qualitative methods to identify the bounds and features of reproducibility discourse, and identify distinct vocabularies and constituencies that reformers should engage with to promote change.

## Introduction

A unique characteristic of recent conversations about rigor, reproducibility, and replication is that they are a truly transdisciplinary phenomenon, not confined to any single scientific discipline. A 2016 survey in *Nature* found that a majority of scientists across a wide range of disciplines had personal experience of failing to reproduce a result, and that a majority of these same scientists believed that science was presently facing a "significant" reproducibility crisis [1]. Reproducibility conversations are also unique compared to other methodological

definition for each thematic code and illustrative examples) are available at are available at: https://github.com/nicole-c-nelson/reproducibility-CA. The NVivo file containing the coded articles is available on request. All code used for analysis is available at: https://github.com/nicole-c-nelson/reproducibility-CA.

**Funding:** NCN received financial support for the Radcliffe Institute for Advanced Study in the form of a Residential Fellowship (no grant/award number, https://www.radcliffe.harvard.edu/fellowship-program), and KI and JC received financial support through the Radcliffe Research Partnership Program at the Radcliffe Institute for Advanced Study (no grant/award number, https://www.radcliffe.harvard.edu/fellowship-program/radcliffe-research-partnership). The funders had no role in study design, data collection and analysis, decision to publish, or preparation of the manuscript.

**Competing interests:** The authors have declared that no competing interests exist.

conversations because they have received sustained attention in both the scientific literature and the popular press. Major outlets such as the *Wall Street Journal* [2], the *Economist* [3], and the *Atlantic* [4–6] have all published feature-length articles on reproducibility.

The scale and scope of reproducibility problems, however, means conversations about them appear ill structured. In a news feature accompanying the *Nature* survey, one scientist described the results as "a confusing snapshot of attitudes" which demonstrated that there was "no consensus on what reproducibility is or should be" [7]. Thought leaders in the reproducibility space similarly describe these discussions as unwieldy. In a 2016 perspective article, Steven Goodman, Daniele Fanelli and John Ioannidis of Stanford's Meta-Research Innovation Center at Stanford argued that "the lexicon of reproducibility to date has been multifarious and ill-defined," and that a lack of clarity about the specific types of reproducibility being discussed were an impediment to making progress on these issues [8]. Many commentators have noted that there is considerable confusion between the terms *reproducibility* and *replicability*, and that those terms are often used interchangeably in the literature [9–13]. Victoria Stodden has argued that there are three major types of reproducibility—empirical, statistical, and computational—each of which represents a distinct conversation tied to a different discipline [14].

The attention devoted to reproducibility issues in the popular press adds another dimension of variation. Some commentators have suggested that journalists and the media are responsible for the crisis narrative, translating "rare instances of misconduct or instances of irreproducibility. . .into concerns that science is broken," as a *Science* policy forum article puts it [15]. Content analysis of news media has similarly suggested coverage of reproducibility issues is promoting a "science in crisis" story, raising concerns among scientists about over-generalized media narratives decreasing public trust in science [16].

To date, scholars have attempted to address these concerns by proposing clarifying definitions or typologies to guide discussions. The National Academies' 2019 report on reproducibility [17] notes the problem of terminological confusion and creates a definitional distinction between *reproducibility* and *replicability*—a distinction that aligns with the usage of these terms in the computational sciences, but which is at odds with the more flexible way they are used by major organizations such as the Center for Open Science and the National Institutes of Health [18, 19]. Numerous other attempts have been made by scholars from both the sciences and humanities/social sciences to clarify the terms of the discussion through conceptual analyses of reproducibility and related concepts such as replication, rigor, validity, and robustness [13, 20–23].

We take an empirical approach to systematizing conversations about reproducibility. Rather than developing analytical definitions, we look for underlying patterns of similarity and difference in existing discussions. Our approach to understanding variation in reproducibility conversations is also more expansive than previous approaches. Rather than focusing solely on differences in terminology, we examine differences in how authors tell the *story* of the reproducibility crisis. This approach offers insight not just into what authors refer to when writing about reproducibility, but also why they believe reproducibility is important (or unimportant), how they came to this realization, and what they think should be done about these issues.

Using a mixed-methods approach, we created a curated data set of 353 English-language articles on reproducibility issues in biomedicine and psychology (Figs 1 and 2) and analyzed the thematic components of each article's narrative of the reproducibility crisis. We hand-coded the articles for four themes: what the authors saw as 1) the *signs* that there is a reproducibility crisis (e.g. a high profile failure to replicate, or an action taken by the NIH), 2) the *sources* of the crisis (e.g. poorly standardized reagents, or misaligned incentives in academic research), 3) the *solutions* to the crisis (e.g. greater transparency in data and methodology, or

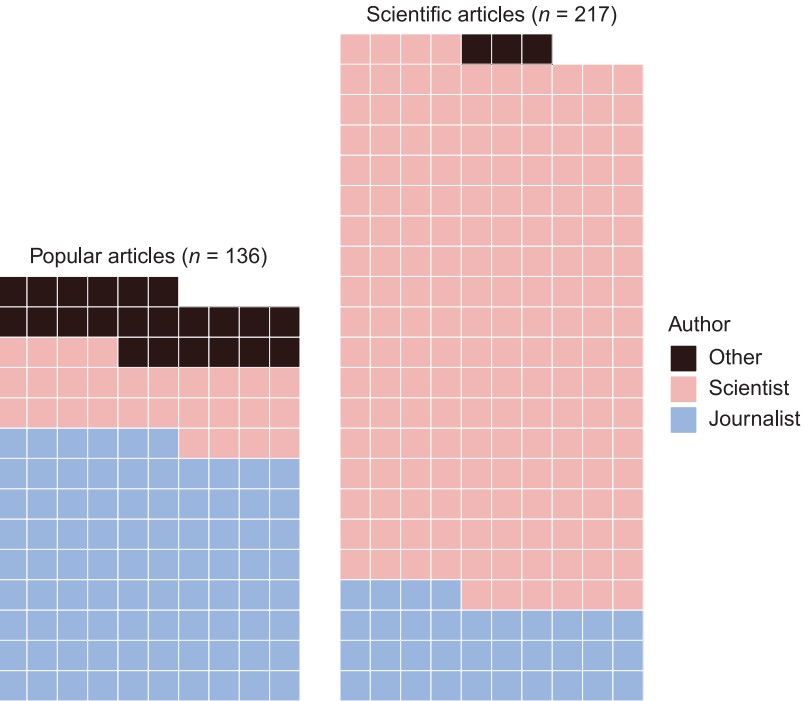

**Fig 1. Data set by audience and author type.** Each block represents one article.

increased training in methods), and 4) the *stakes* of the crisis (e.g. public loss of confidence in science, or the potential for public policy to be built on faulty foundations). The combination of themes discussed and amount of text devoted to each theme creates a unique narrative profile for each article, which can then be compared to the mean article profile for the data set as a whole.

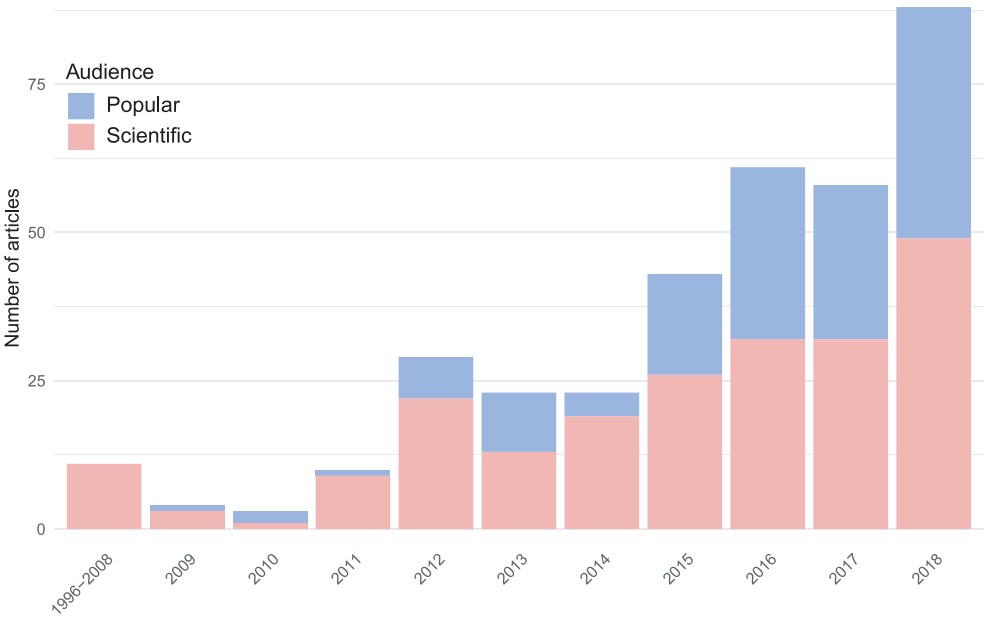

**Fig 2. Data set by year of publication and audience.**

Given that those at the center of reproducibility discussions experience those discussions as ill-structured, we expected to find distinct clusters of discourse: for example, a group of popular articles focusing on fraud as the source of irreproducible results, another group of scientific articles focusing on misuses of null hypothesis significance testing and proposals to change how *p*-values are reported and interpreted, and so on. Instead, we found that the majority of articles in our data set shared a common narrative structure which identified a lack of transparency, misaligned incentives, and problems with the culture of academic publishing as the core causes of irreproducible research.

## Materials and methods

Qualitative research methods are rarely used explicitly in metascience, but they hold great value for understanding the subjective perceptions of scientists. Qualitative research is typically exploratory rather than confirmatory, and uses iterative, non-probabilistic methods for data collection and analysis. The many "researcher degrees of freedom" [24] inherent to qualitative research may raise concerns for readers more well-versed in quantitative paradigms, and can lead to misinterpretations about the conclusions that can be drawn from a qualitative data set. For the present study, the data collection and analysis methods were chosen to allow us to characterize the range and variability of the discursive landscape. However, they do not allow for conclusions to be drawn regarding the relative prevalence of different themes or narratives in a larger body of reproducibility discussions.

### Data collection

We collected English-language articles discussing reproducibility issues using a maximum variation sampling strategy. Nonrandom, purposive sampling strategies such as maximum variation sampling are common in qualitative research because they yield "information rich" cases that are most productive for answering the research question at hand [25–27]. In the present case, maximizing variation increases the chances of identifying rare narratives that might be difficult to see in a random sample, while at the same time allowing us to identify shared patterns that cut across cases. If reproducibility discussions are ill structured or consist of distinct clusters of conversation, maximum variation sampling aids in characterizing the full extent of that variation. If reproducibility conversations are homogeneous, maximum variation sampling allows us to draw even stronger conclusions than a random sample would—any "signal [that emerges] from all the static of heterogeneity," as Michael Quinn Patton puts it, is of "particular interest and value in capturing the core experiences and central, shared dimensions of a setting or phenomenon" [25].

We employed an iterative version of maximum variation sampling: We first collected a sample of articles that maximized variation along dimensions suggested by the existing literature. We then analyzed that sample to identify rare article types, and finally collected a second sample to maximize those rare article types. Based on existing commentaries about the structure of reproducibility conversations, we aimed to maximize variation in 1) discipline, 2) terminology, and 3) audience. We chose to focus on biomedicine and psychology, since reproducibility discussions have been especially active in these fields and have generated substantial popular press coverage (compared to fields such as computer science, where reproducibility issues have been extensively discussed by scientists but not the popular press). We used two databases specializing in scientific literature (Web of Science, PubMed), and two databases specializing in print mass media (Nexis Uni, ProQuest). Using multiple databases introduces redundancy that can compensate for the potential weaknesses of each individual database. For

example, not all subfields of psychology may be equally well represented in PubMed, but may be better captured in Web of Science since it includes humanities and social science indexes.

To maximize heterogeneity in terminology, we used multiple search strings with variations and combinations of the following terms: "reproducibility," "irreproducibility," "credibility," "confidence," or "replicability;" "translational research," "medical," "clinical," or "research;" "crisis," "problem," or "issue." For each query, we reviewed the results and collected articles relevant to reproducibility, excluding articles on clearly unrelated topics (e.g. DNA replication). When search strings retrieved many relevant results (e.g. >500) and could not be feasibly narrowed by modifying the search string, we purposively sampled the relevant articles by selecting rare article types (e.g. non-US articles, articles in smaller journals or newspapers, blog posts). We stopped searching when new permutations of searches revealed few novel articles.

In the resulting data set, the following types of articles were rare: articles published before 2014, articles published in online venues and aimed at popular audiences (e.g. blog posts, online magazines such as *Slate*), non-US articles, political opinion articles, conference proceedings, and white papers from professional societies. To further maximize heterogeneity in our data set, we searched specifically for those rare article types by: 1) following links/citations to rare article types within the articles we had already collected, 2) searching for earlier publications by the authors already identified, and 3) searching for media coverage of key events that took place prior to 2014. It should be noted that these search strategies may have minimized variation along some dimensions while maximizing variation along others. For example, searching for earlier publications by the authors identified in the first round maximized variation in year of publication, but likely did not increase the number of unique authors included.

This search process began in September 2018 and was completed in December 2018, and resulted in a total data set of 464 articles. This number may seem small, but it is in line with prior research: One study using similar search strings in Web of Science found only 99 articles that the author identified as discussing the reproducibility crisis (searching across all scientific disciplines and publication years ranging from 1933 to 2017) [28]. We refined our data set to select for "information-rich" articles. In searching for articles that described irreproducibility as a "crisis," "problem," or "issue," we aimed to identify articles that included a narrative recounting of the reproducibility crisis as a scientific/intellectual movement [29], rather than merely describing a method for enhancing the reproducibility or validity of a specific technique. We excluded research articles that did not contain such a narrative, as well as hearing transcripts and press releases, since these genres did not routinely include a reproducibility narrative. Our final data set contained 353 articles. Complete bibliographic information for the data set (including the articles excluded from the analysis) is available at: https://www.zotero.org/groups/2532824/reproducibility-ca.

## Qualitative data analysis

We employed grounded theory methodology [30] to develop a coding scheme to analyze the themes present in the data set. Grounded theory methodology is widely used in qualitative research to derive new theory inductively from empirical data. It is especially useful when little is known about the phenomena under study, or when existing theories do not adequately capture or explain the phenomenon. It proceeds in two phases. The "open coding" phase involves a process of generating and iteratively refining "codes" that capture particular themes in the data. This is followed by a "focused coding" phase where the entire data set is then re-coded using the coding scheme generated during the first phase. While grounded theory methodology asks researchers to pay close attention to the themes as they are expressed by the people

under study, it also recognizes that all researchers carry "conceptual baggage" [31] which influences their interpretation of the empirical material. The three authors (N.C.N., K.I., J.C.) who developed the coding scheme each have different disciplinary backgrounds and began with different degrees of familiarity with reproducibility discussions: K.I. and J.C. were undergraduate students with training in neuroscience and anthropology, respectively, and had relatively little familiarity with qualitative data analysis or reproducibility discussions on beginning the project. N.C.N.'s training is in the field of science and technology studies, and she began the project with an extensive background in qualitative data analysis and a moderate degree of familiarity with reproducibility discussions.

To make use of the differing perspectives embodied in our research team, N.C.N., K.I., and J.C. each independently selected a random sample of 15–25 articles from our data set and created a list of the themes present in those articles. We compared our lists, noting common themes and generating broader umbrella themes that summarized more specific themes. We refined these thematic codes until 1) our codes covered most of the themes that could reasonably arise in our articles, and 2) each code was specific enough to reveal meaningful trends in code frequency patterns, but general enough to apply to common ideas across articles. To maximize inter-rater reliability (IRR) in applying the codes to the data set, we then performed three rounds of code refinement. All three coders independently coded a group of articles randomly selected from our data set, and we compared our applications of the coding scheme to the articles and revised the coding scheme to improve consistency.

We then coded all 353 articles afresh using this refined coding scheme. Articles were assigned to each coder using a random number generator, and each then independently coded her assigned articles using the qualitative data analysis software NVivo12 [32]. In cases where a sentence contained more than one theme, we coded that sentence with both themes in most cases (S1 Appendix provides a complete description of instances where we opted not to "double code" passages with more than one theme). For the purposes of later calculating IRR scores, fifty-three of these articles (randomly selected but distributed evenly throughout the data set, to account for potential drift in our application of the codes over time) were assigned to and coded by all three coders. In addition to coding the articles for thematic content, we hand-coded the intended audience of each article, the author type, and the main term used in the article (reproducibility, replication, or another term such as "credibility crisis"). The full coding scheme, including the description of each code and examples of the type of discussion included under that code, is available as an appendix (S1 Appendix).

When coding was complete, we merged the NVivo files and conducted pairwise comparisons of IRR on the 53 articles coded by all three researchers using NVivo's "Coding Comparison" query at the paragraph level. Achieving strong IRR scores is a common difficulty in qualitative research, particularly as the codes increase in conceptual complexity and for themes with a low prevalence in the data set [33]. This is reflected in our average Kappa scores (S1 Table): we achieved excellent agreement on codes relating to specific individuals or events (e.g., discussions of John Ioannidis's work or *Nature's* 2016 reader survey on reproducibility), but much lower agreement on codes describing more complex ideas (e.g. that scientists need to change their expectations about what degree or type of reproducibility should be expected). We modified several codes with poor Kappa scores by combining codes that overlapped or by narrowing their scope (details about the specific modifications made are described in S1 Appendix). Only codes that reached an average Kappa score of 0.60 or higher, indicating moderate to substantial agreement between raters [33, 34], were used in the next stage of analysis.

## Correspondence analysis

To visualize similarities and differences between articles and authors, we chose correspondence analysis (CA), a dimensionality reduction technique akin to principal component analysis (PCA) but applied to contingency tables and hence appropriate for categorical and count data [35]. CA also has a long history of use as part of mixed methods approaches for answering sociological questions. Social theorist Pierre Bourdieu famously used multiple correspondence analysis to study the structure of fields and social spaces beginning in the 1970s [36]. CA also has a history of use in examining differences in authors' writing styles, and author data cases frequently appear as canonical examples in CA textbooks and methodological discussions [35, 37]

CA is particularly suited for our analysis because, unlike PCA, it normalizes by the length of a text. Existing analyses show that shorter and longer texts by the same author will appear to be stylistically different when analyzed through PCA, but not through CA [37]. This is important because our data set contains articles ranging from less than one page to more than a hundred, and authors also devote different amounts of space within each article to discussing the reproducibility crisis (e.g. a newspaper article may focus entirely on the crisis, while a longer academic article may only devote a few pages to the subject). CA normalizes the amount of text tagged with each code against the total amount of text coded in the article as a whole, thereby only considering relative proportions within parts of an article that are coded.

We exported tables from NVivo summarizing the percentage of text coded for each theme in each article as.xlsx spreadsheets using the "Summary View" option in the Node Summary pane (available in NVivo for Windows only), as well as tables summarizing article metadata and IRR. We compiled these files into a data frame and performed correspondence analysis using the FactoMineR package (version 2.3) [38]. The 29 codes reaching the Kappa score threshold of 0.60 were treated as active variables in the CA, and word frequency variables and metadata were treated as supplementary variables. Based on a scree plot of the variance explained per dimension, we chose to interpret the first three dimensions of the CA. The data files exported from NVivo and the code for the analysis are available at are available at: https://github.com/nicole-c-nelson/reproducibility-CA.

## Word frequency variables

In CA, supplementary variables are frequently used to assist in the qualitative interpretation of the meaning of the dimensions. These variables do not participate in the construction of the dimensions but can be correlated with the dimensions after they are constructed. We used NVivo's "Text Search" query function to construct several word frequency variables by counting mentions the following terms (and stemmed/related terms) and expressing those word counts as a percentage of the total words in the article: "Gelman", "Ioannidis", "NIH", "psychology", "questionable research practices", and "reagent/antibody/cell line". We used the "Gelman", "Ioannidis", and "reagent/antibody/cell line" variables as an internal double check on our analysis, comparing the position of those word frequency variables to the position of the *Andrew Gelman*, *John Ioannidis*, and *reagent* variables assessed through qualitative data analysis. We also used word frequency variables to aid in the interpretation of Dimension 1. We selected "NIH" as a term that might be more strongly associated with biomedicine, and "psychology" and "questionable research practices" as terms that might be more strongly associated with psychology to assess whether Dimension 1 could be interpreted as capturing disciplinary differences.

## Hierarchical clustering

We used the FactoMineR package [38] to perform hierarchical clustering on the articles in our data set using Euclidean distance. While the distance between rows in the latent dimensions

are $\chi^2$ distances, the projection is onto an orthonormal basis and so it is appropriate to cluster using Euclidean distance [39]. To eliminate noise and obtain a more stable clustering, we retained only the first 18 dimensions from the CA (representing ~75% of the variance). We chose to cluster the data into four classes based on measurements of the loss of between-class inertia (the *inertia* is a measure of variance appropriate for categorical data, defined as the weighted average of the squared $\chi^2$-distances between the articles' profiles and the average profile).

## Bootstrap analysis

Bootstrapping has been applied to CA, but only to capture the variability of the relationship between rows and columns: in our case, that would correspond to uncertainty from ambiguity in coding [40]. But given the importance of maximum variation sampling to our argument, the uncertainty we are interested in quantifying is that resulting from our choice of articles. To simulate this, we constructed 1,000 bootstrap samples by resampling with replacement from the set of 353 articles. These cannot be analyzed with individual correspondence analyses, as the resulting coordinates would not be comparable (CA solutions can be reflected and still be equivalent, and some rotation or scaling might make for more fair comparison). We experimented with Procrustes analysis, but in order to also make the inertia of each sample comparable arrived at Multiple Factor Analysis [41] as a more principled framework [42] for carrying out the bootstrap analysis. Multiple factor analysis (MFA) allows the analyst to subdivide a matrix into groups. This allows the analyst to compare, for example, how groups of people differ in in their responses to survey questions or their evaluations of a quality of an object (e.g., how experts and consumers rate the sensory qualities of the same wines) [42]. In our case, we used MFA to analyze how our 29 themes were positioned in 1) each of the 1000 individual bootstrap samples, 2) our original sample, and 3) all 1001 samples considered together. Using the FactoMineR package [38], we applied MFA to the 353,353 observations of the original and bootstrap samples. To estimate a 95% confidence region in the first factor plane for each theme, we plotted points for each theme's position in each of the 1001 samples, calculated the convex hull around the point clouds for each theme, and calculated peeled convex hulls consisting of 95% of the points, using the method described by Greenacre [35].

## Analysis by author and audience

To examine differences between journalist/scientist and popular/scientific articles, we again used MFA implemented with the FactoMineR package [38]. We grouped the articles using our hand-coded metadata (journalist/scientist author; scientific/popular venue). For the analysis by author type, the groups of articles authored by journalists and by scientists were treated as active (included in the analysis with full mass), and the group of articles by other authors (e.g., members of the general public, employees of policy think tanks) was treated as supplementary (retained in the analysis, but with mass set to zero and hence contributing nothing to inertia). For the analysis by intended audience, the groups of articles (popular and scientific venues) were both treated as active. All code used for analysis is available at: https://github.com/nicole-c-nelson/reproducibility-CA.

## Results

### Correspondence analysis

Correspondence analysis (CA) uses spatial embeddings to visualize relationships between the rows and columns of a matrix. In our case, the unique narrative profile for each article

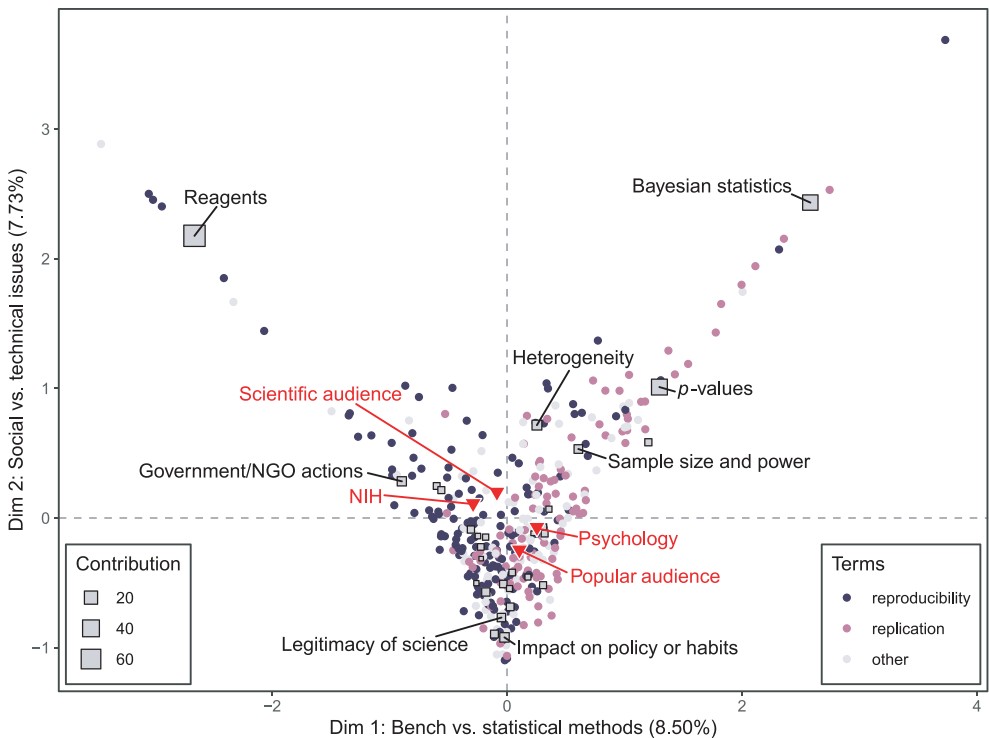

**Fig 3. Correspondence analysis biplot of 353 articles discussing reproducibility, analyzed for 29 themes.** Articles that are close together have similar narrative profiles. The closer an article appears to the center of the plot, the more closely it resembles the mean profile for all articles. The further away a theme is from the origin, the more variation there is in how authors discuss that theme. The color of an article's plotted point (a circle) indicates the main term used in the article, and the size of a theme's plotted point (a square) represents the contribution of that theme to constructing the dimensions. The eight most contributing themes are labeled. Supplementary variables (not used to construct the dimensions) are labeled in red.

(consisting of the amount of text devoted to each theme in that article) was compared to the mean article profile for the data set as a whole. The first factor plane (Fig 3), representing 16% of the variance, captures two distinctions: Dimension 1 separates articles focused on bench work from articles focused on statistical methods, and Dimension 2 separates articles that focus on technical problems from those that focus on the stakes of the crisis.

The distribution of articles across the first factor plane shows many articles clustered around the origin. If reproducibility discourse were composed of distinct clusters of conversation, we would expect to see relatively few articles at the origin—the mean article profile would be a theoretical entity representing the average of several clusters distributed across the first factor plane. Instead, we find that many actual articles resemble the mean article profile in their relative coverage across themes. The proximity of many articles to each other and to the origin of the first factor plane indicates an overall lack of variation in reproducibility discussions. On average, articles in our data set devoted a large percentage of text to discussing *transparency*, *incentives*, and *publishing culture* (9.25%, 8.34%, and 5.82% respectively) but these three themes contributed little to the construction of the first three dimensions of the correspondence analysis (S2 Table). This large presence but low variability suggests that these themes constitute the core of contemporary discussions of reproducibility, with most authors discussing these themes to similar degrees. Given that the data set was constructed to maximize heterogeneity, this is a finding that is generalizable to the discourse as a whole.

Two unique clusters of articles are evident in the upper left and upper right quadrants of the first factor plane. Highly contributing articles located in the upper left extremes include scientific publications on antibody and cell line authentication [43, 44], as well as *Nature News* pieces covering these same issues [45]. In the upper right extremes are articles appearing in *Nature News* and *The New York Times* comparing frequentist and Bayesian statistical approaches, and citing thought leaders in psychology such as Eric-Jan Wagenmakers and Uri Simonsohn [46, 47]. These articles are unique not simply because of the extent to which they discuss reagents and statistical issues, but because they discuss these issues without making strong connections to other themes in reproducibility discourse. For example, the title of *Nature News* feature "Reproducibility crisis: blame it on the antibodies," suggests a singular explanation for reproducibility problems, which is reinforced in the body of the article with a quote asserting that "poorly characterized antibodies probably contribute to the problem more than any other laboratory tool" [45]. While articles in this cluster typically assert that reagents are only one contributor to irreproducibility, they do not discuss other contributors in depth.

An analysis of the supplementary variables correlated with the dimensions suggests that Dimension 1 captures some disciplinary differences and Dimension 2 captures some differences in the intended audience of the article. The word frequency variables "psychology" and "NIH" are, respectively, positively correlated ($r = 0.25$, 95% CI [0.15,0.35], $p = 1.24e-06$) and negatively correlated ($r = -0.29$, 95% CI [-0.38, -0.19], $p = 3.32e-08$) with Dimension 1, meaning that discussions of statistical techniques are more closely associated with mentions of psychology or psychologists, while discussions of reagents are associated with mentions of the NIH. This suggests that Dimension 1 could also be interpreted as separating biomedicine and psychology. However, it should be noted that these correlations are weak and that mentions of "questionable research practices" are not well correlated with Dimension 1 ($r = 0.11$, 95% CI [0.01, 0.21] $p = 3.48e-02$), which is surprising given that this term was coined in the context of discussions of statistical practices in psychology [48].

The intended audience of an article is correlated ($r = 0.34$, 95% CI [0.23, 0.42], $p = 9.07e-11$) with Dimension 2. This indicates that the distinction between articles examining technical problems and those that focus on social problems is related to difference in audience. Articles at the negative extreme of Dimension 2 include newspaper op-eds and short articles that focus on problems in science broadly construed [49–51], typically drawing on empirical examples from several disciplines or using evidence from one field to draw conclusions about another (e.g., claiming that evidence of irreproducibility in biomedicine means that half of all findings about climate change might also be untrue). These articles share a focus on the stakes of the reproducibility crisis, directly questioning the legitimacy of the scientific enterprise or expressing the fear that reproducibility issues may cause others to lose faith in science. As one op ed puts it succinctly, "the house of science seems at present to be in a state of crisis" [49]. These articles support existing analyses arguing that media coverage of the reproducibility crisis may undermine public trust in science, although it is worth noting that a number of the articles at the extreme of this dimension are specific to climate science and authored by individuals associated with conservative think tanks (e.g., the Pacific Research Institute for Public Policy).

Dimension 3 (Fig 4) separates out a cluster of articles focusing on the heterogeneity and intrinsic complexity of the natural world. This cluster includes articles by psychologists and neuroscientists arguing that it should not be surprising that many results fail to replicate because of differences in experimenter gender [52] and the "contextual sensitivity" of many phenomena [53, 54]. It also includes articles on animal experiments which argue that increasing standardization might counterintuitively decrease reproducibility by generating results that are idiosyncratic to a particular laboratory environment [55–57]. The clusters of articles discussing *reagents* and *Bayesian statistics* fall at the opposite extreme of this dimension,

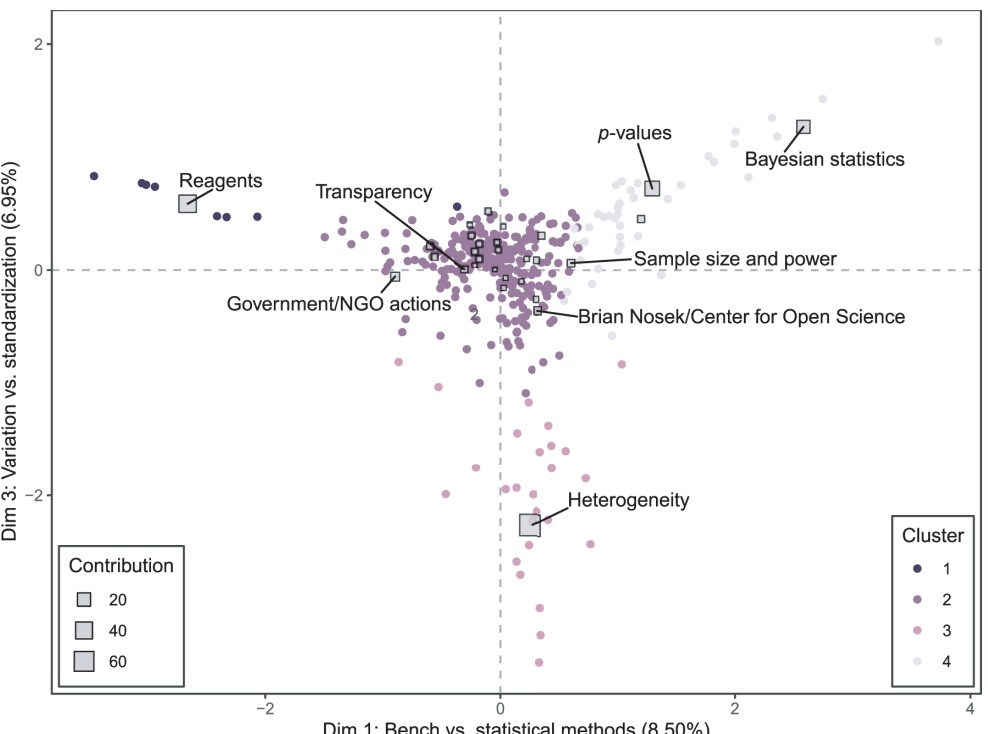

**Fig 4. Correspondence analysis biplot depicting dimensions 1 and 3 of the analysis.** Articles that are close together have similar narrative profiles. The closer an article appears to the center of the plot, the more closely it resembles the mean profile for all articles. The further away a theme is from the origin, the more variation there is in how authors discuss that theme. The color of an article's plotted point (a circle) indicates their label in a hierarchical clustering, based on Euclidean distance in the latent dimensions. The size of a theme's plotted point (a square) represents the contribution of that theme to constructing the dimensions. The eight most contributing themes are labeled.

representing a distinction between authors who tend to focus on standardization and statistics as the solution to reproducibility problems, versus those who see reproducibility problems as the natural consequence of complexity and intentional deployment of variation as the solution. As one article at the negative extreme of Dimension 3 puts it, "experiments conducted under highly standardized conditions may reveal local 'truths' with little external validity," contributing to "spurious and conflicting findings in the literature" [56]. The authors argue that intentional deployment of variation, such as allocating mice to different housing conditions, offers a more promising approach to addressing reproducibility issues than further standardization.

All three of the unique article clusters appearing in Fig 4 (*reagents*, p-*values*/*Bayesian statistics*, and *heterogeneity*) fall into clusters separate from the main body of articles when analyzed using hierarchical clustering. The position of these hierarchical clusters on the factor plane again suggests that reproducibility discourse as a whole is more unified than commentators have assumed. While there are several unique clusters of conversation, the largest cluster is centered at the origin, indicating a low degree of variation in reproducibility narratives.

## Bootstrap analysis

To gain additional insight into themes that cluster together in reproducibility conversations, we conducted bootstrap resampling of our original data set combined with multiple factor analysis (Fig 5). The 95% confidence region for *Bayesian statistics* overlaps with the confidence regions for p-*values* and *sample size and power*, indicating that these three themes cannot be

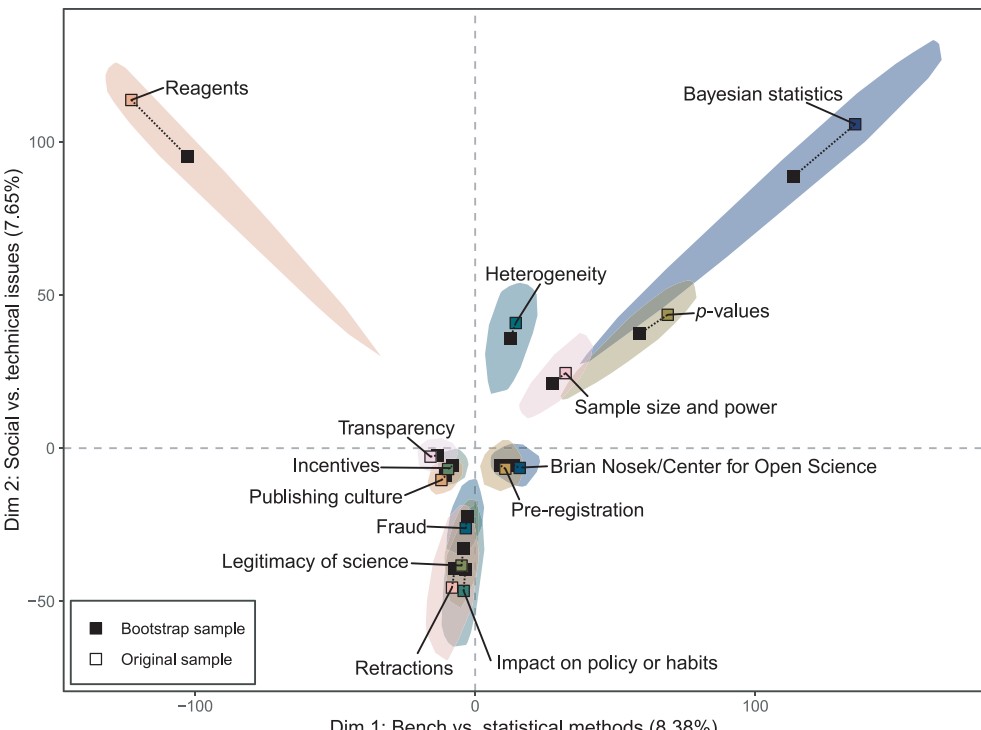

**Fig 5. Multiple factor analysis of bootstrap samples drawn a set of 353 articles on reproducibility.** Shaded areas indicate the peeled convex hulls of points based on 1000 bootstrap replicates plus the original sample, showing an approximate 95% confidence region for each theme. Twenty-nine codes were included in the analysis, but only select codes are displayed for legibility. Colored squares indicate the position of each theme based on the original sample, and black squares indicate the position of each theme based on the overall bootstrap analysis.

reliably distinguished and should be treated as part of a shared discussion about statistical methods. The themes *reagents* and *Bayesian statistics* have elongated confidence regions pointing towards the origin, indicating variation in how these themes are discussed in both of the first two dimensions. While some articles discuss these reagents and statistics to the exclusion of other themes, other articles discuss them alongside other potential sources of irreproducibility. The asymmetry of the bootstrapped peeled convex hulls is because few articles uniquely focus on reagents or statistics; when those articles are not sampled into a particular bootstrap draw, the closer association of those themes with others draws their positions closer to the origin. The overlapping confidence regions for *fraud*, *legitimacy of science*, *retractions*, and *impact on policy or habits* show a similar structure, but with variation primarily in the second dimension. This suggests that the degree to which an article discusses these four themes varies along with the audience of the article but not with the type of science or disciplines it discusses. The core themes *transparency*, *incentives*, and *publishing culture* are overlapping and found near the origin, and also near the origin are *pre-registration* and *Brian Nosek/Center for Open Science*. Although the confidence regions for these two central clusters are distinct in our analysis, this is likely an artifact of our coding scheme (S1 Appendix). When authors discussed the Center for Open Science's efforts to enhance transparency of data and methods, we elected to code these passages as *Brian Nosek/Center for Open Science* and not *transparency*, and this decision is likely responsible for their apparent distinction.

The bootstrap analysis also provides insight into the effect of the maximum variation sampling strategy on the analysis. The position of each theme in the overall bootstrap analysis is

closer to the origin than the position of each theme in the original analysis, indicating that the total inertia (recall: the weighted average of the squared $\chi^2$-distances between the articles' profiles and the average profile, used as a measure of variance) of the bootstrap samples is on average smaller than those of the maximum variation sample. The difference in position between the original and bootstrap analysis is greater for themes with greater variation, such as *reagents* and *Bayesian statistics*. This suggests that the maximum variation sampling strategy has, as expected, captured more variation than would be expected in a random sample or in the population of articles as a whole.

## Analysis by author and audience

Finally, we used multiple factor analysis to compare the profiles of articles authored by journalists versus those by scientists, and articles published in scientist-facing venues versus public-facing venues. When articles authored by journalists and scientists are considered separately, the first factor plane resembles the results of the correspondence analysis performed on the entire data set (Fig 6). The themes with the greatest within-theme inertia on the first factor plane (indicating larger differences in how scientists and journalists discuss these themes) are *Bayesian statistics*, *sample size and power*, p-*values*, and *reagents*. Scientists tend to write more than journalists about these technical issues, although journalists do sometimes devote substantial attention to highly technical topics [46]. Journalists discuss questions about the legitimacy of science and Bayer/Amgen scientists' reports of their experiences of failures to replicate [58, 59] at greater length than scientists. Overall, however, scientists and journalists address similar themes to similar extents when writing about reproducibility.

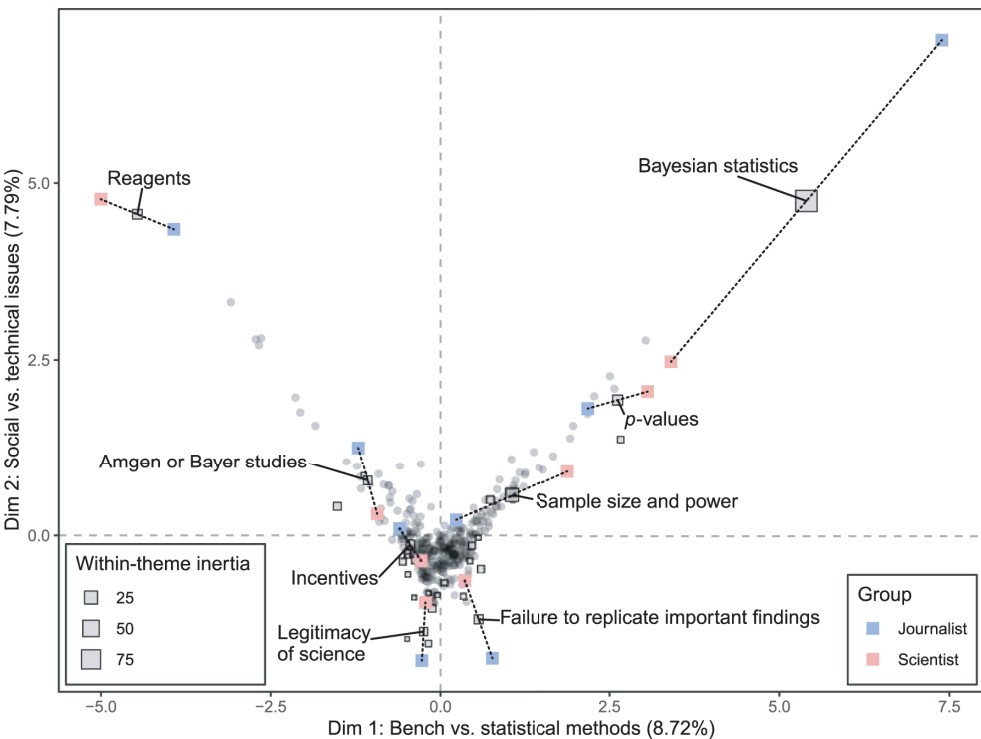

**Fig 6. Multiple factor analysis of 353 articles discussing reproducibility, analyzed for 29 themes, with articles grouped by author type.** The size of the square indicates the degree of within-theme inertia. The eight themes with the greatest within-theme inertia are labeled, and partial points for those themes are displayed. Red points indicate the location of those themes in the group of articles authored by scientists, and blue points indicate the theme location for the journalist group. Grey points indicate the location of the articles on the first factor plane.

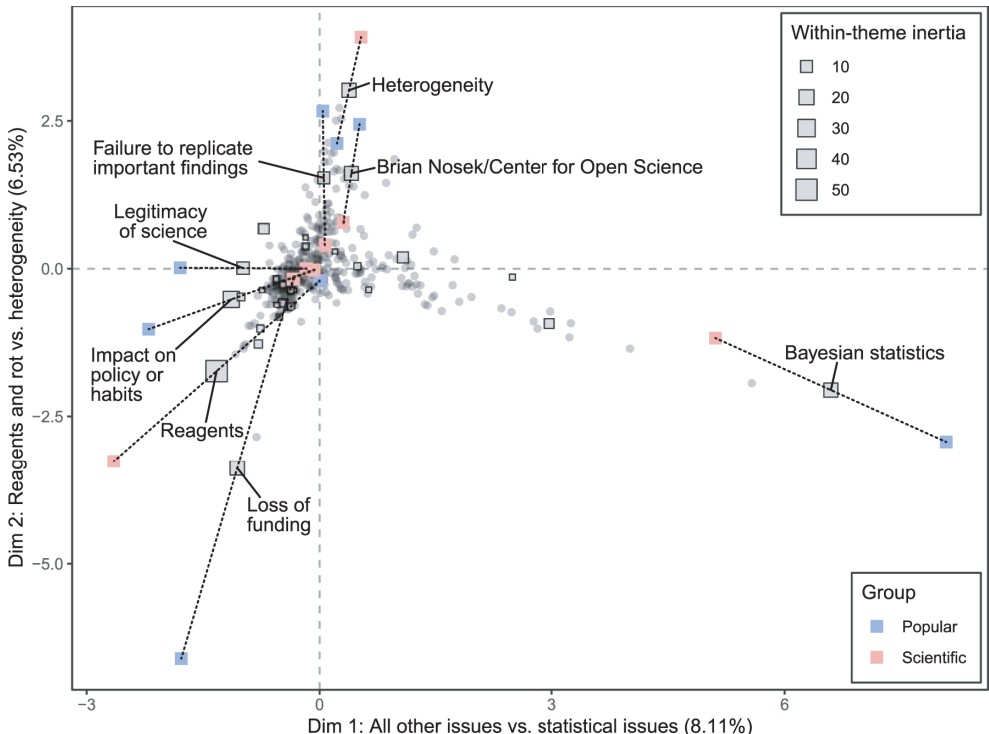

**Fig 7. Multiple factor analysis of 353 articles discussing reproducibility, analyzed for 29 themes, with articles grouped by intended audience.** The size of the square indicates the degree of within-theme inertia. The eight themes with the greatest within-theme inertia are labeled, and partial points for those themes are displayed. Red points indicate the location of those themes in the group of articles aimed at a scientific audience, and blue points indicate the theme location for the popular audience group. Grey points indicate the location of the articles on the first factor plane.

Larger differences are evident when comparing articles written for a scientific audience to those appearing in popular venues (Fig 7). The RV coefficient (a measure of similarity between squared symmetric matrices) for the journalist and scientist author groups is 0.71, while for the scientific and popular article groups it is 0.52. In the analysis by audience, Dimension 1 separates articles focused on statistics from the main body of articles, and Dimension 2 makes visible two narratives that are prominent in popular articles but not in scientific ones. At one extreme of Dimension 2 are articles focusing on heterogeneity, as well as a cluster of popular articles found near the themes *Brian Nosek/Center for Open Science*, *failures to replicate important findings*, and *heterogeneity*. These articles share a narrative of the replication crisis that is specific to psychology, which discusses the failure to replicate long-established findings such as priming effects alongside the COS's efforts to estimate the reproducibility of other findings in the Reproducibility Project: Psychology [60]. Articles authored by journalists in this group argue that there is a crisis in the field [61, 62], and articles authored by psychologists in this group rebut the idea that there is a crisis by appealing to the heterogeneity and complexity of the phenomena psychologists study [53, 54].

At the other extreme of Dimension 2 are articles focusing on reagent issues, as well as a more loosely clustered collection of popular articles that are generally pessimistic about science and scientists. This group includes articles that question whether academic science is really as good at self-correction [63] or at generating novel ideas [64] as it is generally presumed to be. Misconduct, retractions, and the idea that there is "rot" at the core of science are prominent themes [65]. The tone of this cluster is perhaps best encapsulated by a *Fox News* opinion piece

titled "Has science lost its way?," which argues that "the single greatest threat to science right now comes from within its own ranks" [66].

## Discussion

Taken together, our findings suggest that conversations about the reproducibility crisis are far from ill structured. The specific structure we describe here should not be taken as a definitive mapping of reproducibility discourse—it reflects the particular themes included in our analysis, and including additional themes or using a different ontology of themes would result in a different structure. Nevertheless, this model of the reproducibility crisis's discursive structure is useful in that it points towards potential strategies that reformers could adopt to advance reproducibility-oriented change in science.

Our findings suggest that there is a clear thematic core to reproducibility discussions, one that is shared by biomedical scientists and psychologists, and by scientists and journalists. Given that our data set was constructed to maximize heterogeneity, this is an especially noteworthy finding. This core pertains primarily to the sources of irreproducibility and associated solutions; more variation was present in what authors identified as the signs of crisis or the stakes of continued irreproducibility. This is good news for reformers, in that it suggests that consensus is present around the important question of what reforms should look like. Individuals may be convinced by a variety of signs and motivated by a variety of rationales (e.g., the need to reduce costs in drug development, or to preserve the public credibility of science), but these diverse experiences and motives appear to point towards similar interventions.

Our analysis did not identify any strong differences in how journalists and scientists write about the reproducibility crisis. It is worth noting that we did not distinguish between different ways that authors might discuss the same theme (e.g., arguing for or against the need for transparency), and the cluster of articles in Fig 7 discussing the crisis in psychology raises the possibility that journalists and scientists may address the same themes but arrive at different conclusions. However, the supplementary variables correlated with the second dimension of the correspondence analysis and the RV coefficients for the multiple factor analyses both indicate a relationship between the venue in which an article is published and the themes it addresses. This suggests that scientists and journalists alike may foreground different aspects of reproducibility when writing for a popular audience, and therefore both bear some responsibility for popular narratives that call the "well-being" of science into question [16].

It is tempting to attribute the coherence of reproducibility discourse to the emerging discipline of metascience, given the strong resemblance between the core themes identified in this study and the core concerns of this new field. Metascientists have been central players in discussions about reproducibility, and have been especially active in research related to scholarly communication and open science [67]. However, time series analysis shows that the thematic core identified here has been present even from the very early days of reproducibility discussions [68], suggesting that it emerged before or alongside metascience rather than as a result of the formation of the field. The cluster of themes identified here is also present in articles by authors who would be unlikely to self-identify as metascientists, such as Francis Collins and Lawrence Tabak's early paper outlining the NIH's plans for enhancing reproducibility [19].

Although there is a clear core to reproducibility discussions, some elements are better integrated than others. Correspondence analysis, subsequent hierarchical clustering in the latent dimensions, and our bootstrap analysis all indicate that some articles discussing reagents, statistics, and heterogeneity are distinct from the main body of articles we analyzed. Reformers should take note of these minority constituencies in crafting their arguments and interventions, because these single-issue and heterogeneity-focused groups of authors are less likely to

see a need for systemic reform. Our findings suggest that there is a subset of scientists, not confined to any particular discipline, who see the natural world as more intrinsically variable than their colleagues do and are therefore less inclined to see failures to replicate as problematic. We also generally observed in our close reading of the articles (although we did not code for this explicitly) that those authors who saw reproducibility problems as largely attributable to a single factor tended to focus either on antibody specificity or overuse/misuse of *p*-values. While we are not able to draw conclusions about the size of these constituencies based on our analysis, we are able to identify their distinct orientation towards reproducibility problems.

Differences in core assumptions about heterogeneity also explain why distinctions between *direct* and *conceptual* replications remains controversial: to those who see the natural world as deeply variable, few (if any) replications would truly count as direct replications. While those who employ the direct/conceptual distinction acknowledge that it is one of degree rather than of kind, and that not everyone will agree on what counts as a replication [69], they do not acknowledge that individuals appear to vary widely in their baseline assumptions about variation. For reformers, this suggests that attempts to use replication to advance theory development are likely to be frustrated if they do not take into account this diversity in scientists' worldviews.

Articles also differed in their usage of the terms *reproducibility* and *replication*. The primary term used in an article is correlated ($r = 0.38$, 95% CI [0.29, 0.47], $p = 2.87e\text{-}12$) with Dimension 1, suggesting that these terms might be markers of methodological or disciplinary difference. While our maximum variation sampling strategy may have exaggerated the degree of variation in terminology, our analysis points towards the presence of established patterns of use unrelated to the terms' meanings, which may interfere with attempts to create definitional distinctions. This does not mean that there is no value for reformers in making distinctions between types of reproducibility/replication problems; rather, it is to say that hanging distinctions on these two terms seems likely to generate more confusion than clarity.

Finally, this analysis illustrates the value of bringing qualitative approaches to bear on reproducibility. While metascientists have had great success in employing quantitative methods to understand reproducibility problems, these approaches are limited in their ability to identify subjective differences in the meaning individuals give to different terms or events, or to explain why scientists act (or fail to act) in particular ways. Hand-coding is necessary to overcome problems as simple as different words and terms expressing similar ideas (or conversely, the same word or terms expressing different ideas). But the data set produced from this analysis, with high-quality manual annotations for content and themes, will also support research and benchmarking in areas like topic modeling.

Many commentators argue that reproducibility is a social problem that will require changes to the culture of science [70], and yet methodologies designed for studying cultural variation and change—participant observation, ethnography, cross-cultural comparisons, and qualitative data analysis—are only rarely employed in metascientific or reproducibility-oriented research. Achieving lasting change in scientific cultures will first require a more systematic understanding of variation in how scientists interpret reproducibility problems in order to create "culturally competent" interventions.

## Supporting information

**S1 Checklist. SRQR checklist.**
(PDF)

**S1 Table. Inter-rater reliability Kappa scores for all themes coded.**
(PDF)

**S2 Table. Percentage share of the mean article profile, coordinates, contribution, and *cos*$^2$ for all themes included in the correspondence analysis.**
(PDF)

**S1 Appendix. Code book for reproducibility data set.**
(PDF)

## Acknowledgments

We thank Ashley Wong for her assistance with article collection, Harald Kliems for his assistance with producing the figures, and Megan Moreno for providing advice about inter-rater reliability issues. We also thank the Department of Medical History and Bioethics at the University of Wisconsin—Madison for providing publication fee assistance.

## Author Contributions

**Conceptualization:** Nicole C. Nelson, Momin M. Malik.

**Data curation:** Nicole C. Nelson, Kelsey Ichikawa, Julie Chung.

**Formal analysis:** Nicole C. Nelson, Kelsey Ichikawa, Julie Chung.

**Funding acquisition:** Nicole C. Nelson.

**Investigation:** Kelsey Ichikawa, Julie Chung.

**Methodology:** Nicole C. Nelson, Kelsey Ichikawa, Julie Chung.

**Project administration:** Nicole C. Nelson.

**Software:** Nicole C. Nelson, Momin M. Malik.

**Supervision:** Nicole C. Nelson.

**Validation:** Nicole C. Nelson, Kelsey Ichikawa, Julie Chung, Momin M. Malik.

**Visualization:** Nicole C. Nelson, Momin M. Malik.

**Writing – original draft:** Nicole C. Nelson.

**Writing – review & editing:** Nicole C. Nelson, Kelsey Ichikawa, Julie Chung, Momin M. Malik.

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
