## [Decision Letter · Decision Letter 0]

19 Nov 2020

PONE-D-20-26565

Mapping the discursive dimensions of the reproducibility crisis: A mixed methods analysis

PLOS ONE

Dear Dr. Nelson,

Thank you for submitting your manuscript to PLOS ONE. After careful consideration, we feel that it has merit but does not fully meet PLOS ONE’s publication criteria as it currently stands. Therefore, we invite you to submit a revised version of the manuscript that addresses the points raised during the review process.

As you can see below, the reviewers found the topic under study very interesting. Nevertheless, they all raised concerns about methodological choices made and their justification in the text. Considering PLOS ONE's publication criterion #3 (https://journals.plos.org/plosone/s/criteria-for-publication#loc-3), these comments must be properly addressed when revising the manuscript. Please notice that, when presenting mixed-methods studies, we recommend that authors use the COREQ checklist, or other relevant checklists listed by the Equator Network, such as the SRQR, to ensure complete reporting (http://journals.plos.org/plosone/s/submission-guidelines#loc-qualitative-research).

Moreover, Reviewer 3 made some comments on the results (more specifically, about the interpretation of Dimensions 1 & 2). This should also be addressed, as the journals publication criterion #4 reads "Conclusions are presented in an appropriate fashion and are supported by the data.".

We look forward to receiving your revised manuscript.

Kind regards,

Sergi Lozano

Academic Editor

PLOS ONE

Journal Requirements:

3. We noted in your submission details that a portion of your manuscript may have been presented or published elsewhere:

'One related manuscript is under consideration for a special issue of the journal _General Review of Psychology_, to be published in 2021. The manuscript under review is based on the same data set but analyses this data using a different technique (subset correspondence analysis), answers a different question (how have reproducibility conversations changed over time?), and is intended for a different audience (historians and philosophers of psychology).'

Please clarify whether this publication was peer-reviewed and formally published.

If this work was previously peer-reviewed and published, in the cover letter please provide the reason that this work does not constitute dual publication and should be included in the current manuscript.

Reviewers' comments:

Reviewer's Responses to Questions

**Comments to the Author**

1. Is the manuscript technically sound, and do the data support the conclusions?

Reviewer #1: Yes

Reviewer #2: No

Reviewer #3: Yes

2. Has the statistical analysis been performed appropriately and rigorously? 

Reviewer #1: I Don't Know

Reviewer #2: No

Reviewer #3: Yes

3. Have the authors made all data underlying the findings in their manuscript fully available?

Reviewer #1: Yes

Reviewer #2: Yes

Reviewer #3: Yes

4. Is the manuscript presented in an intelligible fashion and written in standard English?

Reviewer #1: Yes

Reviewer #2: Yes

Reviewer #3: Yes

5. Review Comments to the Author

Reviewer #1: In ”Mapping the discursive dimensions of the reproducibility crisis: A mixed methods analysis” the authors analyze a large set of writings about reproducibility in Biomedicine and Psychology. They sample writings on this topic from a broad set of texts, both scientific and popular. They then manually code the content and themes of each article and report the dispersion of content across the most important dimensions. If articles are thematically similar they group together. While initially believing that the discourse of reproducibility is sprawling and badly coordinated, the results of the paper seem to indicate the opposite. Indeed, most articles are thematically close, even though the study includes research from both social and natural sciences.

I thank the authors for a very interesting read. Having worked with reproducibility and open science for a long time, but never with qualitative methods I see the submitted work as part of a very welcome and important methodological development of research into open and reproducible science. While I lack deep knowledge of the methods used, it seems to me the study fulfils the requirements for publication in PLOS One. I have a number of comments mostly related to the structure of the paper and some suggestions for additional analyses which I believe would make the results and conclusions more clear.

Since I am not experienced in the methodology I will in this report focus on the other aspects of the paper. My main comment relates to the motivation and current structure of the paper. The authors could do a better job at outlining exactly what it is they do and what their results are. Currently, the connection between the motivation and the results that are being presented is weak. The abstract and introduction gives the impression that the main contribution of the paper is the mapping of topics, but at least after the fact most topics seem obvious. Instead, the discussion section, which I found very interesting, seems to focus on the finding that despite sampling to maximize heterogeneity, most papers group closely together. This seems to me like the most important finding of the paper and is something I think the authors should write much more clearly in the abstract and introduction. But the importance of these results completely depends on what one would expect. The authors claim in the introduction that the discourse is ill-structured, but this is not motivated by any previous research or even quotations. It seems important to know why the authors thought so at the outset? And especially, how such dispersion would look in terms of results. As a reader, I have little ability to benchmark the results. How much dispersion would we want to see to say that the discussion is ill-structured? An analysis with papers in the same field but on different topics could help here, but is of course a considerable endeavor.

I am uncertain why the authors decided to focus on biomedicine and psychology. Is it specifically because the fields are different? It becomes a bit strange when they write about themes around “reagents” and Bayesian statistics are outside of the core of the discussion. Indeed, these topics are mechanically distinct since they are field-specific. But they could (and likely are) at the core of the discussion within each field. It is not obvious to me that we should expect, or even want, a complete coordination of discourse between fields as different as biomedicine and psychology. Perhaps it would be worthwhile to run some analysis on the topics separately, or to compare psychology to the discussion in other social sciences. This could also be useful as a benchmark.

I wonder if it could be that the authors have in fact identified a distinct discipline. That the reason why papers group so closely on this topic across two such different fields as biomedicine and psychology is because they are more connected across fields than within their own respective field. It would be really interesting if the authors could discuss this and perhaps show some results. It could very well be a problem for the reproducibility movement if it is too closed off from its different fields. That everyone who is working with reproducibility speaks the same language is good, but not if it means that they cannot communicate with the rest of the researchers in their own fields.

Short comments:

As a paper using mixed methods and about scientific discourse, I miss quotations and examples from the included texts. How different can two very different papers be? And what are examples of surprisingly similar articles?

The authors sample to maximize heterogeneity in an attempt to “stack the cards against them”. Finding that articles are closely grouped together thematically is more surprising the more heterogeneous the sample is. This is great but not explained clearly enough!

Reviewer #2: This paper makes use of some of the methodological tendencies of the metascience movement-- namely, statistical analyses of variables derived from word searches of an arbitrarily selected body of texts-- in order to purportedly make a contribution to the growing literature on the problems of reproducibility/replication in modern science. Much of my reaction is grounded in an attempt to understand the larger motivations of the metascience movement growing out of science studies; and here I would encourage the authors to consult the paper “metascience as a scientific social movement” by David Peterson and Aaron Panofsky at UCLA. This will provide some background for my current comments.

There is a long history of work done on the relative scarcity of replication and its variants in science, especially dating back to the work of Harry Collins in the 1980s. This literature explained why replication was difficult, and how scientists dealt with this obstacle in their own research. The problems were there approached both as a matter of practical methodology and of profound philosophical complications, particularly as science studies held as a tenet that there did not exist any unique transdisciplinary ‘scientific method’. Starting after the turn of the millennium, there grew up a movement that ignored all that previous work, but began positing the existence of a putative distinct contemporary crisis of reproducibility (particularly in biomedicine and psychology) that could be rectified by application of the monolithic “scientific method” to the problem; dubbed henceforth as ‘metascience’. One characteristic attribute of the latter was to become especially focused on techniques of formal inductive inference, and allied dependence upon Big Data and computational considerations. At this point a number of semantic shifts were evoked to differentiate ‘reproducibility’ from replicability, itself a topic strewn with pitfalls. This turn received benediction from the National Academies volume on Reproducibility and Replicability in Science (2019).

There is much about the metascience movement to give one pause; Peterson and Panofsky do a good job of raising the issues. Yet, whatever one’s attitudes towards metascience as such, the authors of the current paper push the practices one step further, resulting in an exercise that lacks credibility. They in effect propose a meta-metascience exercise, where they code the appearance of certain words in a set of articles appearing in arbitrarily chosen databases (described on pp.5-6) that they have decided bear upon discussions of problems of reproducibility, and then subject them to statistical analysis of variance, all with the avowed purpose of clarification of the state of discussion of replicability within science. This leads the tendencies of metascience to bedlam, since statistical/computational exercises on their own applied to methodological/philosophical discussions of the nature of science (as opposed to actual empirical scientific research) can do almost nothing to clarify what are inherently abstract disputes over the nature of science and the conditions of its success or failure.

Since there is a looming danger of misunderstanding, let me rephrase this objection. Statistical analysis of variance of what an arbitrary crowd of authors say about replicability cannot serve to clarify what are essentially sociological and philosophical problems of research behavior and organization. (This is doubly compromised by the authors renouncing any interest in the one quantitative aspect of their coding exercise, insisting they are unconcerned about ‘how much’ their authors talked about reproducibility on p.8.) The very notion that there is an abstract ‘scientific method’ that can be applied to long-standing disputes over the nature and conduct of scientific research is itself a category mistake, although it is a hallmark of the tendency of metascience to instrumentalize what are rather deep-seated organizational and conceptual problems besetting modern scientific research. Scientists impatient with philosophical discussion of replication and the social organization of science will simply repeat the errors of their forebears.

The essential irrelevance of the authors’ statistical exercise is itself illustrated by the relatively jejune and pedestrian ‘findings’ they claim in this paper: that some papers focus on individual difficulties of replication such as lack of transparency and variance of inputs to the experimental apparatus (all long long ago covered in Collins), or that “there are no strong difference in how scientists and journalists write about the reproducibility crisis.” These could readily be developed and recounted in any competent survey article, without all the superfluous rigmarole of ‘correspondence analysis’ and the like. The only reason for this mountain to bring forth such an unprepossessing mouse is that the paper wants to assume the trappings of ‘scientific quantitative analysis’ when that recourse is lacking in any real substance or intrinsic motivation. Treating statistics as a self-sufficient ‘method’ is how we got to the modern impasse in the first place.

One might conclude by suggesting this complaint is sometimes broached by science studies scholars concerning the metascience movement in general; I merely mention this to suggest this review is written from within a particular perspective, which is not represented in any of the citations to this article.

For these reasons, I cannot support the publication of this article.

Reviewer #3: Referee report on PONE-D-20-26565

Mapping the discursive dimensions of the reproducibility crisis:

A mixed methods analysis

Using a mixed methods approach, the authors identify the discursive dimensions in the discussion about reproducibility in science. In particular, the authors conduct correspondence and multi-factor analyses on 350 articles, analyzed for 30 themes, to address the question whether discussions about the reproducibility crisis systematically differ along several dimensions like the type of authors (scientists vs. journalists) and the type of the intended audience (popular vs. scientific outlets). The paper identifies the incentive structure of science, transparency of methods, and the academic publishing culture as the main themes of discussions about reproducibility. The authors report systematic differences in the discursive structure depending on whether the article addresses a popular or scientific audience, but relatively little differences in how scientists and journalists write about reproducibility issues. Despite the scale and scope of the crisis, the results suggest that discussions about reproducibility follow a relatively clear underlying structure.

First, I want to admit that I am not at all an expert on the analysis methods applied by the authors. Apparently, this should be considered when reading my comments on the manuscript. As I would expect that a large share of the potential audience of the paper are not too familiar with details of the methods either, I hope that my comments help the authors to revise the paper in such a way that it is more accessible to readers.

Second, I want to emphasize that I very much sympathize with the research question addressed and that I acknowledge the contribution to the literature. <from experience="" in="" my="" own="">

Overall, the paper is well written, convincingly motivated, and clearly structured. I think the paper is novel in terms of the research question addressed and might be of interest to a broad audience. However, the description of the methods is a bit sparse in some aspects and lacks arguments for particular design choices (see below for details and examples). Moreover, for an audience who is not familiar with the methods used (as I am), some claims and interpretations do not seem to follow “naturally” from the results presented in the paper. Below, I outline my main concerns in some detail and hope that my comments will help to improve the paper further.

Comments on methods and materials:

Data collection:

The authors collected articles from various databases using combinations of different search strings. The results were reviewed by the authors and only “articles relevant to reproducibility” (p. 5) were included in the sample. “To further maximize heterogeneity,” the authors handpicked additional articles that were rare in the article type. This process apparently involves ample degrees of freedom. The authors address neither how they came up with the selection of the databases nor how they generated the list of search strings.

Skimming the bibliographic information for the data set provided via Zotero further raises some questions. First, some key contributions to the discussion on replicability are not included in the sample (e.g., the Many-Labs studies, etc.). In more general terms, I wonder how the list of search strings used translates into the rather small sample of only 350 articles. Given the scale and scope of the discussion across the social and natural sciences in the scientific literature and the popular press, I would have expected a much larger number of articles. Second, and potentially more relevant to the methodology, several authors appear repeatedly (e.g., Aschenwald, Baker, Engber, Gelman, Ioannidis, Lowe, Nosek, Yong, etc.), and articles in the sample are not balanced across relevant dimensions of the analysis (type of authors, type of audience, year; Fig. 1 and 2). How does this fit the intend to maximize variability in the sample?

While I understand that jointly maximizing variation across various dimensions is a non-trivial task, I wonder according to which criteria the sample was actually constructed. Given the rather vague description of the sampling strategy and the various degrees of freedom in the data collection, to me the final sample appears as being generated by a black box. Providing the interested reader with a more thorough description of the data collection process (perhaps in a detailed section in the appendix) could considerably improve the paper.

Coding Scheme:

Related to the previous comment, I consider it difficult for readers to get a clear idea of how the authors derived the thematic codes for their analysis. The authors’ approach involves a seemingly infinite number of degrees of freedom. For instance, the authors counted mentions of a particular strings (e.g., “Gelman,” “psychology,” etc.; p. 7) but it is not at all clear what is the rational for counting exactly these words.

Another issue that puzzles me with respect to the coding is that several of the codes ended up with very low average interrater reliability scores. Terms that appear highly relevant to the analysis – such as, e.g., “Replication,” “Failures to replicate,” “Effect size,” or “Selective reporting” – have average Kappa values way below 0.6. Moreover, several codes that I would consider important (e.g., “harking”/”forking,” “open science,” “publication bias,” “researcher degrees of freedom,” “transparency,” etc.) do not at all appear of the list. I think the methods section, overall, could be improved by describing the derivation of the coding scheme more thoroughly (potentially in a separate section in an annex to the paper).

Overall, given the topic addressed and the concerns raised in the comments above, it is a pity that the methodology used in the paper was not pre-registered. Yet, I commend the authors for transparently sharing their data and analysis scripts.

Comments on the results:

Overall, I commend the authors for presenting the results in a way that is accessible even to readers like me, i.e., someone without detailed knowledge about the methods used. The presentation and discussion of the results are easy to follow and present novel insights into the research question. The figures neatly depict the main findings and the figure captions are clear and self-contained (a small exception is Fig. 5 and 6 where the labels of the dimensions are missing).

Indeed, from my point of view, there is quite little to criticize about the results section. A minor point is that Dimension 1 is initially referred to as separating articles focused on bench work from studies on statistical methods, whereas Dimension 2 is defined as separating articles focusing on technical issues from articles focusing on the stakes of the crisis. In the following, however, the authors state that “Dimension 1 can be interpreted more broadly as representing disciplinary differences between biomedicine and psychology/statistics, and Dimension 2 can be interpreted as representing differences in the intended audience of the article.” While I agree that this broader definition seems to be supported, I do not see this interpretation clearly follows from the by the analysis. Other minor points that puzzles me – such as the choice of themes and supplementary variables, for instance – are likely due to the issues addressed in my comments on the methodology.

One more minor point: The discussion of Fig. 6 only refers to Dimension 2 but does not address Dimension 1. Although the pattern of clusters along Dimension 1 appears to be similar to the pattern identified in Fig. 5, it might be worthwhile to address the variation along Dimension 1 for the intended audience too (for the sake of completeness).

However, given the concerns discussed in my first comments on the degrees of freedom in the data collection and data analysis stage, a thought that comes immediately to my mind is how robust the results are to different samples and/or subsamples. Personally, I think that robustness analysis (e.g., on bootstrapped subsamples) could be a worthwhile exercise to rule out that the main findings are not driven by the sampling strategy, for instance. Another related issue is the question how generalizable the results are. Given the sampling strategy aiming at maximum heterogeneity in the sample, the articles are likely not representative for its population. The discussion section, however, tends to generalize the patterns identified in the sample – justifiably so?

An aspect not addressed by the analysis is the time dimension. Apparently, it would be interesting to see whether differences in the use of words and terms used in articles along the various dimensions analyzed are correlated with the year in which the article appeared. Put differently: did the patterns of discursive dimensions systematically change over time? I think this question arises quite naturally and addressing it could add an interesting result to the paper.</from>

6. PLOS authors have the option to publish the peer review history of their article (what does this mean?). If published, this will include your full peer review and any attached files.

Reviewer #1: No

Reviewer #2: No

Reviewer #3: No

---

## [Author Response · Author response to Decision Letter 0]

17 May 2021

Editor:

1. Complete COREQ or equivalent reporting checklist. A completed SRQR checklist has been submitted with the revised manuscript. The COREQ checklist is formatted for interview-based qualitative studies, and so we opted to use the SRQR checklist instead.

2. Address formatting issues. We have re-generated the bibliography in PLOS ONE format.

3. Explain why some data is not publicly available. We have made the data and code needed to reproduce the analyses in the manuscript available on GitHub. This includes the outputs of the qualitative data analysis performed using NVivo, in spreadsheet format. We have not included the original NVivo file since this file contains the full text pdfs of the articles we analyzed and making this publicly available may constitute a copyright violation.

4. Clarify the status of the Review of General Psychology manuscript. The related manuscript has been accepted with minor revisions at RGP but has not yet been published. We have added a citation to this forthcoming paper in the revised text. As stated previously in the submission letter, the RGP manuscript is based on the same data set but analyses this data using a different technique (subset correspondence analysis), answers a different research question (how have reproducibility conversations changed over time?), and is intended for a different audience (historians and philosophers of psychology). It does not constitute dual publication because very little text is shared between the two articles, apart from the methodological description of the sample collection and coding process.

Reviewer One:

1. The initial assumption that reproducibility conversations are ill-structured is not supported by references to existing research. We have added multiple references to the introduction to better support this claim.

2. It is difficult for readers unfamiliar with CA to “benchmark” the results. The point is well-taken that the correspondence analysis is less meaningful without some background expectation, or what the CA would look like under greater heterogeneity. It is difficult to provide quantitative benchmarks since the numeric scales produced through CA cannot be meaningfully compared across data sets, but we have added additional text at the beginning of the results section that describes qualitatively what our results may have looked like if reproducibility discourse was more heterogeneous. 

3. The rationale for focusing on biomedicine and psychology is not clear. We have added additional text to the Materials and Methods/Data collection section to explain that we selected biomedicine and psychology because reproducibility issues in these fields have been discussed extensively both in the scientific community and the popular press.

4. The findings may suggest that reproducibility research might constitute a distinct discipline of its own. This is an interesting suggestion, which we now take up in the revised conclusion. We now argue that while the presence of a shared discourse may be partially explained by the emergence of metascience as a discipline, but not fully, since the shared discourse extends beyond the community of authors who self-identify as metascientists.

5. Include quotations and examples from the texts analyzed. We appreciate this invitation to expand the presentation of our results in this direction and have included more representative quotes throughout the results section.

6. Provide a clearer explanation of how the sampling strategy employed is useful for addressing the research question posed. We appreciate the feedback on our description of the sampling strategy. We have added additional text describing why purposive sampling strategies are common in qualitative research, and why maximum variation sampling is especially appropriate for characterizing the range and variability of reproducibility narratives. 

Reviewer Two:

1. The manuscript conducts statistical analyses of variables derived from word searches of an arbitrarily selected body of texts. We regret that we failed to convey to this reviewer key pieces of our methodological approach. First, the frequencies are not based on word searches, but on a qualitative coding scheme that we have now described in greater detail, both in the Methods section and in the S1 Appendix. Second, the texts are not arbitrarily selected, but have been selected through maximum variation sampling. We have expanded our description of this sampling strategy, as described above in R1.6.

2. The analysis “statistical/computational exercises on their own applied to methodological/philosophical discussions of the nature of science... can do almost nothing to clarify what are inherently abstract disputes over the nature of science.” We cannot answer the question of whether computational techniques alone (e.g., topic modeling/natural language processing) can be useful in clarifying reproducibility discussions, since this is not the approach that we employed here. Our mixed methods approach uses qualitative techniques that are appropriate for parsing the different positions taken by actors within a social world, in combination with quantitative approaches to visualize the results of that analysis. We suspect that this concern is rooted in the misunderstandings identified in R2.1 above. 

3. The decision to normalize the counts of coded text using the total amount of coded text (rather than using raw counts or normalizing the counts against the total length of the text) is inappropriate. We have added additional text to explain why this choice is appropriate. As explained in the Abdi and Williams (2010) article that we cite, using PCA in our case would fail to reveal meaningful similarities in style between authors because of the differences in the length and format of the texts.

4. The findings presented in the manuscript could be derived through alternative methods such as systematic review, and the use of correspondence analysis is “lacking in any real substance or intrinsic motivation.” We agree that it may be possible to arrive at similar results using alternative methods, but it does not follow that there is no value to the methods we use here. Given the large number of texts, themes, and metadata variables we analyzed, exploratory data analysis techniques are helpful for summarizing and visualizing the main features of the data set. We believe that the results of our analysis are more succinctly presented through visualizations in combination with narrative than they would be through a narrative format alone.

Reviewer Three:

1. The manuscript needs a more thorough description of the article selection method, including rationales for the selection of databases, selection of search strings, and how maximizing variation along some dimensions (e.g. year of publication) may reduce variation in others (e.g. the appearance of multiple articles by the same authors in the data set). We have added more detail on the rationales for selecting the fields studies, databases employed, and search strings used to the methods section (see also R1.3). We have also added a note to the methods section acknowledging that maximizing variation in some areas may have reduced variation in others. 

2. The number of articles is smaller than expected given the scale and scope of reproducibility discussions. We have added a citation to Fanelli (2018), which describes a bibliographic data set generated using search terms that are comparable to ours, and which identifies only 99 publications related to the reproducibility crisis.

3. The process for deriving the thematic codes is not clear and involves many “degrees of freedom.” Some expected codes (e.g., “harking”/”forking,” “open science”) do not appear on the list. We have added additional text to the methods section to describe the process of code development through grounded theory methodology, including the important distinction between the “open coding” phase where codes are generated, and the “focused coding” phase where established codes are applied to the body of texts as a whole. We hope that this alleviates some of the concerns about researcher degrees of freedom, although we also note in the revised methods text that some of these concerns may arise from differences inherent to quantitative and qualitative research paradigms. We have also incorporated the code book (originally provided as a standalone document on the GitHub repository) as a new appendix to the paper (S1 Appendix). The code book describes in detail what was included under each code, which will help readers identify how themes of interest were parsed according to our scheme (e.g., discussions of open science would be classified under “transparency of data and methods” in our analysis).

4. The rationale for performing the text search queries is not clear. We have added additional text to the Materials and Methods section to clarify the purpose of creating the word frequency variables, which serve as a check on the qualitative analysis and to enhance the interpretation of the CA dimensions.

5. The Kappa values of many thematic codes appears unexpectedly low and raises concerns about the exclusion of these codes from the analysis. Low Kappa scores are very common in qualitative research, particularly as codes increase in conceptual complexity and for codes that appear at low frequency in the corpus. We have added additional text in the Materials and Methods/Qualitative data analysis section to clarify why it is difficult to achieve strong Kappa scores. We have also added text to the discussion section to explain that the CA mapping presented here should not be interpreted as a definitive mapping, since the configuring of the map would shift depending on the codes included (or if a different coding scheme altogether were used).

6. The methodology was not pre-registered. We agree that this is a limitation. At the time we began this study, we were not aware of any options for pre-registration that were tailored to qualitative research. The OSF, for example, released its first template for pre-registration of qualitative research in September 2018, very shortly after we began our study. To compensate for this limitation, we have attempted to be as transparent as possible in reporting our methods and data.

7. The rationale for the expanded interpretation of Dimensions 1 and 2 is not clear. We are grateful for the encouragement to revisit our interpretations here because in rechecking our analysis we discovered an error in the r value originally reported for the “psychology” word frequency variable, which suggests that our original interpretation that Dimension 1 was also capturing differences in discipline was not as sound as we had initially thought. Accordingly, we have revised the manuscript so that Dim 1 is presented as separating bench vs. statistical techniques and Dim 2 is interpreted as separating social vs. technical issues. We still discuss the relationship of these two dimensions to discipline and audience, but are now more careful in our presentation of the potential weaknesses of these interpretations. 

8. The dimensions have not been labeled in Figures 5 and 6. A discussion of Dimension 1 in Figure 6 should be included for completeness. We have corrected these omissions.

9. A robustness/bootstrap analysis would provide insight into how the sampling method has impacted the results. We thank the reviewer for this generative suggestion. We have implemented a bootstrap analysis by using the same multiple factor analysis design used to analyze articles by author and by audience. This analysis does demonstrate the impact of the sampling strategy on the results, but it has also allowed us to identify additional clusters of themes at the core of the conversation that can be differentiated with confidence. The results of this new analysis are presented in Fig 5 of the revised text.

10. It is unclear whether the results of the analysis can be generalized to the population as a whole. We have added additional description about the sampling strategy (see also R1.6) as well as a general caveat to the beginning of the methods sections about what kinds of conclusions can be generalized to the population as a whole, and what kinds of conclusions cannot be drawn from our analysis. We have also added several caveats to the discussion section to remind readers of these limitations.

11. The manuscript could explore how the discursive dimensions might have changed over time. We address this question in a separate manuscript: the forthcoming Review of General Psychology manuscript described in point Editor.4 above.

---

## [Decision Letter · Decision Letter 1]

21 Jun 2021

Mapping the discursive dimensions of the reproducibility crisis: A mixed methods analysis

PONE-D-20-26565R1

Dear Dr. Nelson,

We’re pleased to inform you that your manuscript has been judged scientifically suitable for publication and will be formally accepted for publication once it meets all outstanding technical requirements.

Kind regards,

Sergi Lozano

Academic Editor

PLOS ONE

Additional Editor Comments (optional):

Reviewers' comments:

Reviewer's Responses to Questions

**Comments to the Author**

1. If the authors have adequately addressed your comments raised in a previous round of review and you feel that this manuscript is now acceptable for publication, you may indicate that here to bypass the “Comments to the Author” section, enter your conflict of interest statement in the “Confidential to Editor” section, and submit your "Accept" recommendation.

Reviewer #1: All comments have been addressed

Reviewer #3: All comments have been addressed

2. Is the manuscript technically sound, and do the data support the conclusions?

Reviewer #1: Yes

Reviewer #3: Yes

3. Has the statistical analysis been performed appropriately and rigorously? 

Reviewer #1: Yes

Reviewer #3: Yes

4. Have the authors made all data underlying the findings in their manuscript fully available?

Reviewer #1: Yes

Reviewer #3: Yes

5. Is the manuscript presented in an intelligible fashion and written in standard English?

Reviewer #1: Yes

Reviewer #3: Yes

6. Review Comments to the Author

Reviewer #1: I thank the authors for addressing my thoughts and comments and for the interesting read. I am happy to recommend the paper for publication.

Reviewer #3: All comments have been properly addressed and the manuscript has been considerably improved. Particularly, the description of the methodology is way clearer now.

7. PLOS authors have the option to publish the peer review history of their article (what does this mean?). If published, this will include your full peer review and any attached files.

Reviewer #1: No

Reviewer #3: No

---

## [Editor Report · Acceptance letter]

1 Jul 2021

PONE-D-20-26565R1 

Mapping the discursive dimensions of the reproducibility crisis: A mixed methods analysis 

Dear Dr. Nelson:

I'm pleased to inform you that your manuscript has been deemed suitable for publication in PLOS ONE. Congratulations! Your manuscript is now with our production department. 

Kind regards, 

on behalf of

Dr. Sergi Lozano 

Academic Editor

PLOS ONE